# Genome Characterization and Spaciotemporal Dispersal Analysis of Bagaza Virus Detected in Portugal, 2021

**DOI:** 10.3390/pathogens12020150

**Published:** 2023-01-17

**Authors:** Marta Falcão, Margarida Barros, Margarida D. Duarte, Fábio Abade dos Santos, Teresa Fagulha, Margarida Henriques, Fernanda Ramos, Ana Duarte, Tiago Luís, Ricardo Parreira, Sílvia C. Barros

**Affiliations:** 1Institute of Hygiene and Tropical Medicine (IHMT), NOVA University of Lisbon, Rua da Junqueira 100, 1349-008 Lisbon, Portugal; 2Faculdade de Ciências Médicas, NMS|FCM, NOVA Medical School, Universidade Nova de Lisboa, Campo dos Mártires da Pátria 130, 1169-056 Lisbon, Portugal; 3National Institute of Agrarian and Veterinarian Research (INIAV, I.P.), Av. da República, Quinta do Marquês, 2780-157 Oeiras, Portugal; 4Centre for Interdisciplinary Research in Animal Health (CIISA), Faculdade de Medicina Veterinária, Universidade de Lisboa, Avenida da Universidade Técnica, 1300-477 Lisbon, Portugal; 5Global Health and Tropical Medicine (GHTM), Rua da Junqueira 100, 1349-008 Lisbon, Portugal

**Keywords:** *Flavivirus*, BAGV, genome, *Alectoris rufa*, phylogenetic analysis, spaciotemporal analyses, Israel turkey meningoencephalomyelitis virus (ITV), Ntaya group

## Abstract

In September 2021, Bagaza virus (BAGV), a member of the Ntaya group from the *Flavivirus* genus, was detected for the first time in Portugal, in the heart and the brain of a red-legged partridge found dead in a hunting ground in Serpa (Alentejo region; southern Portugal). Here we report the genomic characterization of the full-length sequence of the BAGV detected (BAGV/PT/2021), including phylogenetic reconstructions and spaciotemporal analyses. Phylogenies inferred from nucleotide sequence alignments, complemented with the analysis of amino acid alignments, indicated that the BAGV strain from Portugal is closely related to BAGV strains previously detected in Spain, suggesting a common ancestor that seems to have arrived in the Iberia Peninsula in the late 1990s to early 2000s. In addition, our findings support previous observations that BAGV and Israel turkey meningoencephalitis virus (ITV) belong to the same viral species.

## 1. Introduction

The genus *Flavivirus* within the Flaviviridae family constitutes a group of over 84 known enveloped viruses, that includes both *bona fide* arboviruses and insect-specific viruses. Like other flaviviruses, Bagaza virus (BAGV) has a positive sense, single-stranded RNA (ssRNA +) genome of approximately 11 kb in length. It includes a single uninterrupted open reading frame (ORF) that encodes a large polyprotein precursor, which is co- and post-translationally processed by cellular and viral proteases into at least 10 distinct proteins. The 5′ region of the viral ORF encodes three structural proteins: the capsid €, the pre-membrane (prM), which is post-translationally cleaved to produce pr and M protein, and the envelo (E) proteins. The E coding sequence is followed by those encoding the so-called non-structural proteins (NS1, NS2A, NS2B, NS3, NS4A, NS4B, and NS5 (RNA-dependent RNA polymerase/methyltransferase)). The single ORF is flanked by two untranslated regions (UTR), the 5′ of which possesses a methylated cap that facilitates translation and protects the viral genome from degradation, while the 3′ UTR, which forms complex stem-loop secondary structures, also facilitates translation and replication [1].

Within the *Flavivirus* genus, Bagaza virus (BAGV) is a member of the Ntaya group, along with five other species, namely Ntaya virus (NTAV), Ilheus virus (ILHV), Tembusu virus (TMUV), Rocio virus (ROCV), and Israel turkey meningoencephalomyelitis virus (ITV). BAGV shares high sequence similarity with ITV, which was isolated for the first time in 1958 from domesticated turkeys in Israel and later, in 1978, from South African turkeys [2,3]. Due to their high nucleotide sequence identity, both viruses have been suggested to belong to the same species [3]. However, as this suggestion has not yet been made official, both viruses remain as different species (www.ictv.global/report/2laviviridae; accessed 12 December 2022).

BAGV was first isolated from a pool of *Culex* mosquitoes in 1966, in the Bagaza district of the Central African Republic [4]. Subsequently, it has been detected in mosquitoes (*Cx. Perexiguus*, *Cx. Univittatus*) collected in other African countries [5,6], India [7], and the United Arab Emirates [8]. In vertebrates, BAGV-associated infections were first reported in red-legged partridges (*Alectoris rufa*), ring-necked pheasants (*Phasianus colchicu*), and common pigeons in Spain in 2010 [9], and a few years later, in 2016, in Himalayan monal pheasants (*Lophophorus impejanus*) from South Africa [10]. More recently, in 2019, another outbreak was detected in Spain in red-legged partridges coinfected with *Plasmodium* sp. [11].

Mortality due to the 2010 and 2019 BAGV Spanish outbreaks had a serious impact not only on the abundance of natural populations of this species but also on the ecosystem of the Iberian Peninsula [12]. As a native wild gamebird, red-legged partridges have high socio-economic relevance in Portugal, Spain, France, and Italy, being the only autochthone partridge species in Portugal. In the last decades, wild populations have declined due to the deterioration of their natural habitat, the increase in hunting pressure, and the emergence of diseases caused by agents such as BAGV 2021 [13,14], to the point that *Alectoris rufa* was declared as a near-threatened species by the European Union and has been classified as Species of European Conservation Concern since 2021 [15].

A study from 2015 showed that BAGV can be transmitted by direct contact between red-legged partridges [16]. In addition, due to the long-lasting viral loads detected in the calami of immature feathers, it was suggested that feather sampling could be a useful matrix for early detection of BAGV in red-legged partridges in active surveillance programs [16].

Typical symptoms of BAGV in susceptible bird species include weight loss, weakness, and apathy with mortality rates as high as 30–40% in partridges, and lower in pheasants and in common wood pigeons (*Columba palumbus*) [9,12,17]. The disease induces neurologic symptoms in red-legged partridges and ring-necked pheasants and, to a lesser extent, in common wood pigeons [17]. In red-legged partridges, infection by BAGV causes severe hemosiderosis in the liver and spleen that is absent in pheasants and less evident in common wood pigeons. The tropism of BAGV for endothelial cells of different organs has also been demonstrated as being dependent on the species affected. In addition to central nervous lesions common in the three species, in red-legged partridges, a severe hemolytic process has also been reported [17].

The zoonotic potential of BAGV has been suggested based on the detection of anti-BAGV neutralizing antibodies in human patients with meningoencephalitis in India. It has also been demonstrated that BAGV strains circulating in West Africa have codon adaptation to human house-keeping genes [18]. Regardless of this evidence, more studies need to be conducted to clarify the true zoonotic potential of BAGV.

Despite sharing land borders with Spain, where the disease was reported in 2010 and 2019, no evidence of BAGV circulation had been reported in Portugal prior to 2021 [19]. In this report, we carried out the genetic characterization and phylogenetic analysis of the full-length polyprotein genomic sequence of BAGV responsible for the BAGV Portugal 2021 outbreak. To date, few reports have focused on BAGV detection and its molecular epidemiology in birds.

## 2. Materials and Methods

### 2.1. The Study

In October 2021, the carcasses of two red-legged partridges, found dead in a hunting ground in Serpa, in southern Portugal, were received at the National Reference Laboratory (NRL) of Portugal, INIAV I.P., and submitted for necropsy and molecular diagnosis [19]. Several tissue samples from both specimens (including, brain, heart, kidney, spleen, and intestine) were screened for the presence of BAGV by RT-qPCR targeting the NS5 coding region [20]. Tissue samples from both partridges (referred to as no. 7 and no. 8 in Queirós et al., 2022 [19]) revealed the presence of BAGV, and one of these samples (no. 7, which showed the lowest Ct value) was used for further molecular characterization and virus isolation.

### 2.2. RNA Extraction and Viral RNA Detection

For nucleic acid extraction, samples of heart and brain of partridge no. 7 were homogenized at 20% (*w*/*v*) with Phosphate Buffered Saline (PBS), and clarified at 3000× *g* for 5 min at 4 °C. Total DNA and RNA were extracted from 200 µL of the clarified supernatants, using the IndiMag Pathogen Kit (Indical, Leipzig, Germany) and a King Fisher Flex extractor (ThermoScientific, Waltham, MA, USA), following the manufacturers’ instructions.

### 2.3. Virus Isolation

For virus isolation, Vero cells (ATCC CCL-81) and *Aedes albopictus* clone C6/36 cells (ATCC CRL-1660) were used. Heart and brain samples from partridge no. 7, which showed the lower Ct values (20.4 and 21.52, respectively), were chosen for virus isolation. Tissue samples were homogenized at 20% (*w*/*v*) in PBS containing penicillin, streptomycin and amphotericin B (antibiotic-antimycotic), used according to the manufacturer’s instructions (Gibco, Waltham, MA, USA). Following centrifugation (3000× *g*, 10 min, 4 °C), the supernatant was filtered through a 0.45-µm-pore-size filter (Millipore Express, Darmstad, Germany) and used to inoculate sub-confluent (70%) Vero and C6/36 cells, maintained in Eagle’s medium supplemented with 5% FBS, penicillin, streptomycin, and amphotericin B (antibiotic-antimycotic used at 1:100, Gibco), and 50 µg/mL gentamicin (Gibco), incubated, respectively, at 37 °C and 28 °C in a humidified atmosphere with 5% CO_2_ and observed daily for cytopathic effect (CPE) in a phase-contrast microscope. Three passages were done for both samples.

### 2.4. Viral Genomic Sequencing

Complete genomic sequencing of the virus genome was performed by bidirectional sequencing of RT-PCR fragments amplified. The virus strain was named BAGV/PT/2021. Seventeen primer sets were designed based on an alignment of all BAGV complete genomes available in GenBank database (January 2022) (Table 1). RT-PCR reactions were performed with the AgPath-ID™ One-Step RT-PCR Reagents kit (Applied Biosystems, Foster City, CA, USA). PCR products were purified using NZYGelpure purification kit (NZYTech, Lisbon, Portugal) and Sanger sequenced using the ABI Prism BigDye Terminator v3.1 Cycle sequencing kit on a 3130 Genetic Analyzer (Applied Biosystems, Foster City, CA, USA). Whenever necessary for sequencing confirmation, individual RT-PCR products were sequenced after being cloned into the pCR2.1-TOPO vector (Invitrogen, Carlsbad, CA, USA) according to the manufacturer’s instructions.

### 2.5. Phylogenetic and Spatiotemporal Analyses

Phylogenetic analyses were based on two sequence datasets, including different strains of flavivirus listed in Appendix A. The largest of these datasets included 108 flavivirus near full-length (ORF coding section) sequences from no known vector, insect-specific, as well as tick- and mosquito-borne flaviviruses, while the smaller one comprised 61 sequences of only the so-called “Ntaya group viruses” that include NTAV, ITV, TMUV, and BAGV. The alignments were checked for signs of recombination using RDP4 [21].

The analysis of the largest sequence dataset was based on the construction of multiple alignments of nucleotide (nt) sequences using the iterative G-INS-I method as implemented in MAFFT vs. 7 [22] followed by their edition using GBlocks [23], to remove from the alignment poorly aligned sections while maintaining the ORF codon alignment. The largest and smallest sequence datasets corresponded to 8058 and 10,286 aligned nucleotides positions, respectively. Phylogenetic trees were constructed using both the Maximum Likelihood (ML) optimization criteria [24] and a Bayesian approach [25]. For both these types of analyses, the best fitting evolutionary model used was GTR+Γ+I, as suggested by IQ-TREE [24]. The stability of the obtained ML trees was assessed using both classical bootstrap and SH-aLRT (Shimodaira–Hasegawa approximate likelihood ratio test), with both tests using 1000 resamplings of the original data. On the other hand, the stability of the obtained Bayesian tree is indicated by the posterior probability (pp) values shown at the tree branches. Bootstrap and aLRT values >75% and pp values above 0.8 were considered significant.

Spatiotemporal reconstruction of the continual BAGV/ITV/TMUV dispersal over time and space was based on two independent Markov-Chain Monte Carlo (MCMC) chains run for 100 × 10^6^ generations under a lognormal relaxed molecular clock (the strict molecular clock hypothesis was rejected using the ML Molecular Cock test implemented in MEGA X [26]), a constant sized population and a Cauchy Relaxed Random Walk as coalescent and demographic dispersal priors, respectively. During the MCMC runs, chain convergence and adequate (>200 after 10% burn-in removal) effective sample size were checked using the Tracer software v1.7.1 (http://beast.bio.ed.ac.uk/tracer) accessed on 12 September 2022. The trees were logged on every 10,000th MCMC step, and their final distribution was summarized with TreeAnnotator software v1.8.3 as a maximum clade credibility (MCC) tree, and finally visualized with the FigTree v1.4.2 software (http://tree.bio.ed.ac.uk/software/figtree/) accessed on 12 September 2022. Finally, the BAGV/ITV/TMUV spatiotemporal reconstruction dispersal was visualized on the Spatial Phylogenetic Reconstruction of Evolutionary Dynamics software, version 3 (SpreadD3) [27], based on a custom-made geoJSON world map (https://geojson-maps.ash.ms/*)* accessed on 12 September 2022.

### 2.6. ITV and BAGV Genetic Analyses

Principal coordinate analysis (PCOORD) of viral sequences was based on a multiple sequence alignment of BAGV (*n* = 21), ITV (*n* = 4), NTV (*n* = 2), and TEMBV (*n* = 1) sequences and was carried out at https://www.hiv.lanl.gov/content/sequence/PCOORD/PCOORD.html (accessed on 12 September 2022), where the first 10 dimensions covered over 93% of the cumulative sequence variation.

Unrooted phylogenetic neighbor-net networks were constructed using the SplitsTree software [28] after selecting Kimura’s two-parameter (K2P) model to correct the calculated genetic distances.

Finally, inter as well as intragroup corrected genetic distances (*d*) were calculated using MEGAX software and the K2P formula.

## 3. Results

### 3.1. Virus Isolation

All attempts for virus isolation in Vero and *Aedes albopictus* clone C6/36 cells were unsuccessful.

### 3.2. Viral Genomic Sequencing

The nucleotide sequence obtained in the course of this study was deposited in the GenBank/EMBL/DDBJ databases under accession number LC730845 (Appendix A).

The polyprotein gene of strain BAGV_PT/2021_ is 10,285 nucleotides long, encoding 3427 amino acids and exhibiting very high (>98%) overall amino acid similarity for the different proteins encoded by the BAGV/Spain (HQ644143), with identical NS2b, NS4a, and 2K proteins, and with the lowest similarity registered for the capsid protein. Similar observations are made when BAGV are compared to ITV-encoded proteins, highlighting the high sequence similarity between both viruses (Table 2). When analyzing the polyprotein, 16 unique amino acid substitutions were found in BAGV_PT/2021_ (Table 3). From these, 56% were non-conservative amino acids substitutions occurring throughout the viral ORF (C, E, NS1, NS2a, NS3, NS4b, and NS5).

When the viral genomic untranslated regions (UTR) were analyzed, the BAGV strains from Portugal and Spain displayed identical 5′-UTR (75 aligned nucleotides), and only differed to that of ITV in two positions. On the other hand, nine substitutions were found in the 3′-UTR (435 aligned nucleotides), this number increasing to 25 when the homologous ITV sequence was included in the comparison. Both the NTAV and TMUV sequences presented both 5′ and 3′ UTRs that were significantly more different to those of BAGV than that of ITV (data not shown).

### 3.3. Phylogenetic and Spatiotemporal Analyses

Additional nucleotide sequence characterization of BAGV_PT/2001_ involved phylogenetic, neighbor-net network construction and PCOORD analyses, and resourcing to different datasets. In addition, a continuous spaciotemporal dispersal analysis using a Bayesian approaches was also carried out.

The first and largest of the sequence datasets included 108 sequences comprising complete ORF coding sequences from different flaviviruses. This initial analysis placed all flavivirus sequences together according to their respective flavivirus group, in all cases supported by high (>75%) aLRT/bootstrap values (Figure 1). In this analysis, the Ntaya group, placed within the mosquito-borne flavivirus radiation, cluster was divided into two monophyletic groups, one grouping together TMUV, BAGV, ITV, and NTAV, and the other clustering ILHV and ROCV. In addition, inside the first group, two additional monophyletic groups were observed, one corresponding to the TMUV genetic lineage, and the other with NTAV, ITV, and BAGV. This last monophyletic group divided itself into two branches, one formed by NTAV sequences and the other by ITV and BAGV sequences (Figure 1).

In-depth analysis of the BAGV sequence from Portugal, showed its phylogenetic proximity to the BAGV from Spain, and a higher distancing from BAGV sequences from the Senegal and Central African Republic, with African and Iberian BAGV strains tending to segregate into distinct monophyletic clusters in a phylogenetic tree (west Africa: Senegal/Côte d’Ivoire/Central African Republic versus southern Africa (Namibia/Zambia) and Iberia (Spain/Portugal). However, the shared ancestry of the BAGV lineage is well supported and also includes ITV in a single genetic lineage with a genetic diversity (*d* = 0.049) that is equivalent to that of the well represented (in terms of total number of available sequences) TMUV lineage (*d* = 0.068).

The high identity between BAGV and ITV was further established using both PCOORD and neighbor-net network reconstruction (Appendix A). In both of these cases, BAGV/ITV clustered together with no segregation between BAGV and ITV sequences, clearly showing that BAGV and ITV viruses tightly cluster together, while NTAV and TMUV viruses grouped in two different independent clusters.

To obtain a better resolution of the phylogenetic relationships shared between viruses, a second phylogenetic tree was constructed with a smaller sequence dataset (Appendix A), exclusively comprised of BAGV, ITV, TMUV, and NTAV sequences. As the phylogenetic analysis presented in Figure 1 showed, these sequences clearly formed a strong monophyletic group, indicating unambiguous shared ancestry. To investigate the way these sequences continuously dispersed in time and space, the phylogenetic tree shown in Figure 2a was constructed using a Bayesian framework, as these analyses allow the assessment of the evolutionary relationships among sequences, while also considering their respective sampling dates and geographic locations.

The Bayesian tree (Figure 2a) confirmed the former analysis (Figure 1), also clustering the viruses from the Ntaya serogroup into two monophyletic groups, one containing BAGV/ITV and NTAV and another TMUV. In the BAGV/ITV monophyletic cluster, four groups were identified, namely G1, G2, G3, and G4, separated by low intra genetic distances (Figure 2b). However, when these values were compared between the major groups (BAGV/ITV group, TMUV group and NTAV group), as expected, genetic distance values increased (Figure 2b). The phylogeographic analysis suggested a most recent common ancestor (MRCA) to all flavivirus analyzed to have existed more than 740 years ago, with no clearly defined geographic location since the error associated was too high. From this uncertain MRCA, the genetic lineage grouping ITV, BAGV, and NTAV spread out to Africa (Sudan), whereas in the case of TMUV, the viruses spread eastward toward Thailand (Figure 2a,c). TMUV viruses shared a common ancestor in Thailand 225 years ago, from where these viruses than diverged to Cambodia and China. On the other hand, NTAV diverged from BAGV/ITV around 400 years ago spreading to the Democratic Republic of Congo. Regarding BAGV/ITV, the analysis suggested a common ancestor around 270 years ago in North Sudan, which approximately 160 years ago diverged from North Sudan to Sudan and then to Chad, arriving later at the Iberian Peninsula. Indeed, the dispersal of the BAGV/ITV sequences around the world probably started in eastern Africa, around year 1636, and only reach south Asia more than 200 years later (around the year 1848). More recently, in 1988 and 1953, respectively, these viruses dispersed to the south and west Africa, from where there were two isolated cases of migration from central Africa, namely to the Iberia Peninsula, around 2003, and to India around 1996 (Figure 2c). According to this model, BAGV possibly arrived at the Iberian Peninsula through Spain in the early 2000s, from where it spread to Portugal. However, due to the reduced size of the dataset used, the dates associated to the tree splitting events may shift as new viral sequences become available in the future, as suggested by the analysis of only ITV/BAGV sequences that placed the common ancestor of the Iberian BAGV sequences in the late 1960s (not shown).

## 4. Discussion

In 2021, BAGV was detected for the first time in resident wild birds (red-legged partridges and corn bunting) in southern Portugal [19], and the whole genome sequencing and genetic analyses (including phylogenetic/phylogeographic reconstructions) were undertaken to characterize this BAGV strain. While BAGV had never been detected in Portugal before, not surprisingly, phylogenetic inference and genetic distance calculations based on the analysis of aligned polyprotein coding sequence of 108 public flavivirus ORFs demonstrated the close relationships between BAGV_PT/2021_ and the BAGV strain from the 2010 outbreak in Spain.

While multiple factors, such as massive tourism and global commercial trading and the limited success of mosquito-control programs could have a significant impact in the expansion of these viruses in recent times, the transcontinental spread of flaviviruses has also been a recurrent occurrence often associated with bird migrations [29,30]. However, the detection of BAGV in the Portuguese territory either by infected birds from Spain during 2021 or by bird movements near the border (assuming that the disease may have persisted unnoticed in southern Spain since 2019) also constitute possible explanations regarding the origin of the infection, given the close phylogenetic relationship with the BAGV from Spain [30].

As the analysis of Fernández-Pinero and collaborators [3] previously indicated, our data also support the fact that BAGV and ITV seem to be members of one single viral species in the Ntaya radiation. The other two members of the Ntaya group formed well-separated genetic lineages, with NATV sharing direct common ancestry with BAGV/ITV, whereas TMUV formed the basal clade. Our results are in agreement with the different lines of evidence that indicate that BAGV and ITV are not just closely related viruses but should be considered the same species. Indeed, they do not segregate into virus-specific subclusters in phylogenetic trees, the genetic distances between ITV and BAGV fall within the range of values obtained when only BAGV sequences are compared with one another, and the proteins encoded by both viruses share overall (>95%) high sequence. The same holds true when the 5′ and 3′ UTS are also compared. ITV causes a neuroparalytic disease with symptoms similar to those induced by BAGV, despite the fact that in adult turkeys, ITV may lead to high morbidity and mortality. This virus was previously reported to be synonymous of BAGV; however, until now, they have not been officially classified as members of the same species [2,3]. In addition to the lack of segregation into different monophyletic clusters in phylogenetic trees, and low inter-BAGV/ITV genetic distances, in agreement with previously published results [3] both PCOORD and network reconstruction were again not able to separate BAGV and ITV, strengthening the idea that ITV and BAGV are different strains of the same viral species. Despite being phylogenetically more distant than ITV, NTAV is still closely related to BAGV, sharing one close common ancestor that emerged around 400 years ago in Africa (Sudan as one of the probable locations), which was not surprising since both viruses belong to the Ntaya serogroup.

Only relatively recently have BAGV sequences been made available in the public databases. Indeed, in the dataset used for the phylogeographic analysis here reported, only 18% of the sequences corresponded to viruses detected prior to year 2000, with the oldest one being found in 1943. In addition, the number of sequences available per geographic location is highly variable. Therefore, and not surprisingly, it is possible that the overall scarcity of sequence data, and their sampling heterogeneity (in terms of uneven distribution in time and space) might have influenced the results obtained, leading to errors in the timing and/or geographic dispersal events suggested by the spatiotemporal dispersal analysis, especially regarding the early dispersal events from the MRCA, where the 95% HPD intervals were large. Indeed, the large 95% HDP intervals estimated for the most internal tree nodes, as well as the proposed ages for the MRCA, despite giving an indication of the times of divergence should be interpreted with caution, especially closer to the root of the tree. Furthermore, a restricted analysis consisting exclusively of BAGV/ITV sequences suggested the common ancestor of the viral sequences detected in the Iberian Peninsula may have existed in the late 1960s which pushes back the later dates (early 2000s) indicated by the larger sequence dataset. The analysis of ORF sequences revealed a tree root for the Ntaya group somewhere close to the Middle East (albeit with low statistical support) from where two major viral lineages diverged: BAGV/ITV and NTAV towards Africa and TMUV towards Asia. Curiously, our analysis also suggested a dispersal out-of-Africa of BAGV/ITV in the mid-1990s towards Asia (India), but no further BAGV/ITV sequences have been detected there ever since.

While it is clear that all these viruses share common ancestry, the TMUV virus is the more divergent one. This virus was first isolated from *Culex* mosquitoes in Malaysia, in 1955, and after the first outbreak in China caused by a variant TMUV (Duck egg drop syndrome virus or DEDSV) in 2010 and affecting egg-laying ducks [31], the infection spread across Asia. TMUV causes a significant decrease in egg production and severe neurological disorders in several species of ducks, being responsible for huge economic losses to the duck-producing industry [31]. Just like BAGV, TMUV antibodies and nucleic acids have been detected in humans, suggesting its zoonotic potential [32].

The genus *Flavivirus* includes medically important mosquito-borne viruses that can infect humans and cause morbidity and death, such as zika, dengue (DENV), yellow fever (YFV), West Nile (WNV), St. Louis encephalitis (SLEV), and Japanese encephalitis (JEV). The potential of BAGV to induce disease in humans is still unclear and reports on its zoonotic potential are extremely scarce [7]. However, being a member of the genus *Flavivirus* and given all the biological and medical information already available from other flaviviruses, surveillance of BAGV and associated vectors will allow more effective and rapid control of the virus, mitigating the implications for public and animal health.

## Figures and Tables

**Figure 1 pathogens-12-00150-f001:**
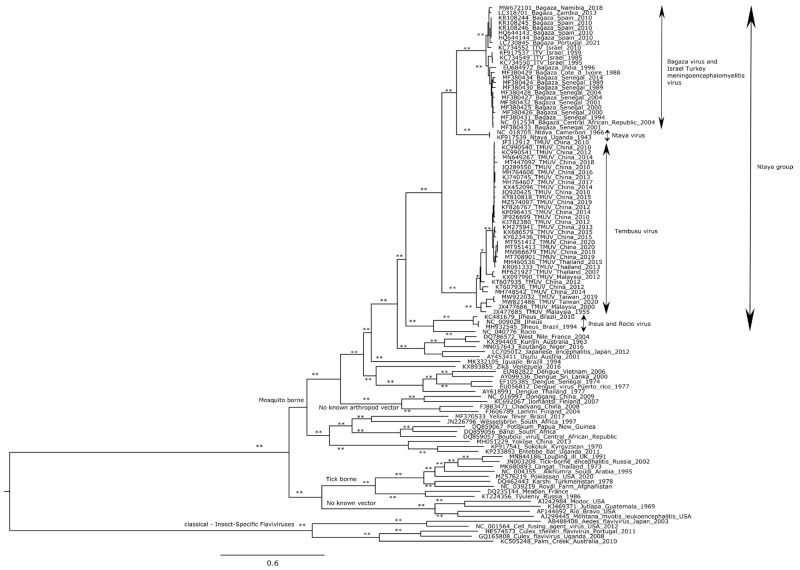
ML tree based on the nucleotide sequences from no known vector flaviviruses, classical insect-specific flaviviruses, as well as tick and mosquito-borne flaviviruses. At specific branches, * indicates those supported by aLRT or bootstrap value >75% (of 1000 data resamplings), whereas ** indicates both aLRT/bootstrap support.

**Figure 2 pathogens-12-00150-f002:**
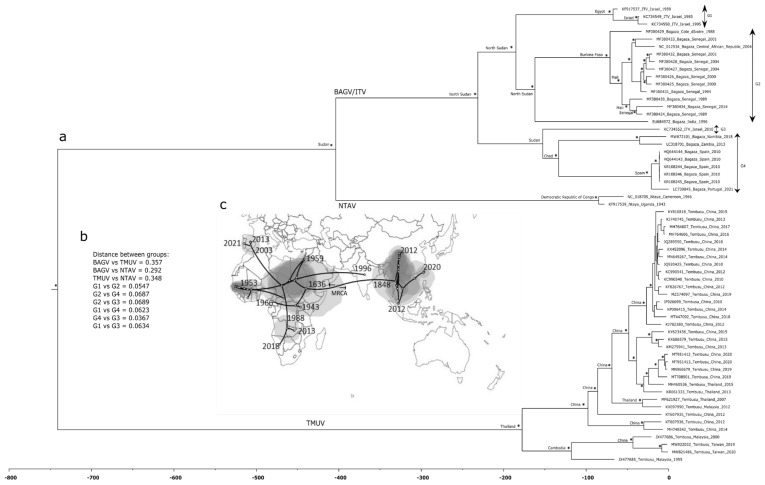
(**a**) Time-scaled Bayesian phylogenetic tree of Ntaya group sequences. At specific branch nodes, posterior probabilities ≥0.90 are displayed by *. The most probable geographic location of the ancestors indicated by the different bifurcating nodes is pointed. The reverse timescale indicates years before the present (data of the most recent sequence). Viral sequences are identified by their name and accession number (underscore); (**b**) intra- as well as inter-group K-2P corrected genetic distances; (**c**) final stage of the putative phylogeographic history of the Ntaya group world spread under uncorrelated lognormal relaxed clock priors, and a continuous diffusion model that estimates the ancestral locations of the viruses in continuous space change using changes in coordinates (latitude and longitude). The circular polygon area is proportional to the number of tree lineages maintaining that location.

**Table 1 pathogens-12-00150-t001:** Primers used for nucleotide sequencing of full-length BAGV polyprotein gene.

Primer	Sequences (5′–3′)	Tm * (°C)	Amplicon Size (bp)
BAGV1F	*ACTTTGTGATTGACAGCTCAA*	53	626
BAGV626R	*GATCATACCCATCCTCTAGCTT*	55	
BAGV516F	*TCCCAACTGCTGGAGGAAA*	58	682
BAGV1197R	*GCTTTGGTGTTGTGGGCCT*	61	
BAGV992F	*GGAGTTGAGTGGATTGATGTT*	54	668
BAGV1659R	*AGTGATTCTCTGTTCTGCC*	53	
BAGV1527F	*GGATGGACATGAGCCAGTTTTA*	56	752
BAGV2278R	*CACTTGGTGTATTCCTTTGC*	53	
BAGV2129F	*CAATGGCACAAGAGTGGAAG*	55	869
BAGV2997R	*ATCACGGCCGTGTCACATTC*	60	
BAGV2771F	*GAGGAATTGGAATACGGGTG*	55	894
BAGV3664R	*CATGTCCCTGTAAGTTATCC*	51	
BAGV3575F	*CGGAAAAGATGGACAGGCCGGG*	65	696
BAGV4271R	*CCATGTCTCCTTCGTCAAAGTGT*	58	
BAGV4081F	*GAAAGGTGGAGTGCTGATTG*	55	740
BAGV4820R	*GTCCTCCATACGATATCAAGTC*	53	
BAGV4688F	*GTAGGAGTGATGTTTGATGG*	51	745
BAGV5432R	*CATCCATCACAAACAAGTTGT*	52	
BAGV5262F	*CAGCTGAGATAGCGGAAGCT*	59	850
BAGV6111R	*GGCTCATAGAGTTGGGCTAC*	57	
BAGV5944F	*GCAGAGACGCGGAAGAATTG*	58	881
BAGV6824R	*TGTCAGTTTGTGATCTCTGTC*	53	
BAGV6725F	*GAAGTCCAACCACAAAAGATAG*	52	924
BAGV7649R	*GCCTCCTCTTCTCATGCTTC*	57	
BAGV7552F	*CCATGTGCCATCTAATGAGGAAG*	57	999
BAGV8550R	*TGTATGGATTTTCTGTGTCAT*	50	
BAGV8313F	*GGAGCCAGTGGGAACATCAC*	60	773
BAGV9085R	*AGCCACATATACCATATGGCCC*	58	
BAGV8988F	*TGTGAGACATGCATCTACAA*	52	721
BAGV9708R	*TGCTGTTCAAGAAATGCAATG*	53	
BAGV9598F	*TGGGCAAGAATGGAAGAGA*	55	839
BAGV10436R	*CTACTTACTTACTTAAATCTGATTA*	45	
BAGV10237F	*CAGAGAACATATACACACCAAT*	50	560
BAGV10796R	*GGGGTCTCCTCTAACCTCTA*	56	

* the indicated Tm values were calculated using the NEB Tm calculator tool (https://tmcalculator.neb.com/; accessed on 12 September 2022).

**Table 2 pathogens-12-00150-t002:** Comparison of BAGV/PT/2021 (LC730845) amino acid sequences with those of other flaviviruses from the Ntaya group.

Proteins	Size ^a^	BAGV/Spain HQ644143	ITV KC734550	NTAV NC018705	TMUV MN649267
		ID%	ID%	ID%	ID%
C	122	96.68	95.08	80.33	70.43
preM	167	99.40	100.00	87.43	82.42
E	501	99.60	98.00	92.02	83.43
NS1	352	99.15	98.01	83.52	77.25
NS2a	227	99.12	97.80	76.21	63.88
NS2b	131	100.00	100.00	85.50	75.57
NS3	619	99.35	98.87	91.44	85.92
NS4a	127	100.00	100.00	91.34	82.68
2K	22	100.00	95.45	86.36	90.91
NS4b	254	98.82	98.82	87.01	83.46
NS5	905	99.67	99.01	90.83	85.86

^a^ Number of amino acid residues; ID% indicates the percentages of amino acid sequence identity; BAGV—Bagaza virus; ITV—Israel turkey meningoencephalomyelitis virus; NTAV—Ntaya virus; TMUV—Tembusu virus.

**Table 3 pathogens-12-00150-t003:** Unique amino acid substitutions in BAGV_PT/2021_ (LC730845). Amino acids positions for mature peptides were annotated based on Ntaya virus (NC018705).

Amino Acid Position	Protein	^1^ ITV/^2^ BAGV_Spain_/^3^ BAGV_P_
83	C	K/K/R
100	C	* G/G/S ^1^
659	E	K/K/R
699	E	* S/S/F ^2^
838	NS1	E/E/D
896	NS1	* W/W/L ^3^
1051	NS1	K/K/R
1202	NS2a	* L/L/M ^4^
1547	NS3	H/H/R
1885	NS3	* Q/Q/P ^5^
1887	NS3	* N/N/Y ^6^
2283	NS4b	* S/S/N ^7^
2358	NS4b	* I/I/T ^1^
2435	NS4b	I/I/V
2703	NS5	I/I/V
3286	NS5	* G/G/S ^1^

^1^ ITV (KC734550); ^2^ BAGV Spain (HQ644143), ^3^ BAGV/PT/2021 (LC730845); * Non-conservative substitutions: ^1^ Aliphatic to hydroxylic; ^2^ Hydroxylic to aromatic; ^3^ Aromatic to aliphatic; ^4^ Aliphatic to sulfur-containing; ^5^ Amidic to aliphatic; ^6^ Amidic to aromatic; ^7^ Hydroxylic to amidic.

## Data Availability

The complete nucleotide sequences of the BAGV sequence detected in this study have been uploaded to the GenBank nucleotide sequence database, accession number LC730845.

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
