# Peer review of "Genome Characterization and Spaciotemporal Dispersal Analysis of Bagaza Virus Detected in Portugal, 2021"

_pathogens, 2023, doi:10.3390/pathogens12020150_

Round 1

Reviewer 1 Report

This manuscript by Marta Falcão et al, aimed to report the Genome Characterization and Spaciotemporal Dispersal Analysis of Bagaza Virus detected in Portugal, 2021. Phylogenies inferred from nucleotide, complemented with the analysis of amino 23 acid alignments, indicated that this strain from Portugal is closely related to BAGV strains previously detected in Spain (2010 and 2019). The authors also argued that there is a crucial need to set up surveillance of BAGV and associated vectors to allow more effective and rapid control of the virus, mitigating the implication on public and animal health.

Please, the following comments should be considered for revisions to the manuscript:

Major revisions

The clarity of the captions should be improved to 300 dpi. They are not readable in the current version.

The comparative analysis performed in this study didn’t include the 5’ and 3’ NCR that play an important role in Bagaza virus diversity. Could the author assess the polymorphism in these regions at least between the BAGV strains from Spain and Portugal?

Lines 304 and 370, the authors have written “probably” which mean that the estimates did not support this statement. Could they provide more details regarding this uncertainty (line 376-380)? I recommend that the authors performed a regressed root-to-tip analysis including only BAGV sequences to ensure sufficient time structures in alignment for reliable rate estimates, determine accurate sampling dates and perform again a separated MCMC for BAGV sequences.

The abstract is very poorly written and could be improved the more details summarizing the findings.

The discussion could be improved by comparing your findings to previous studies.

Minor revisions:

Line 42, could the authors change “the so-called the non-structural proteins” to “the so-called non-structural proteins”?

Line 54, could the authors change “benn made” to “been made”?

Lines 170-171, could the authors rephrase for more clarity?

Lines 214-216, could the authors rephrase for more clarity?

Line 261, could the authors change “lienage” to “lineage”?

Lines 285-291 seems to be a duplicate of a section in the Methods. It could be rephrased or moved to the methods section and avoid mixing methods with results.

Lines 311-318 should be moved before lines 293-310. The authors should describe the MCC from 3a to 3c.

Line 339, change “detected i Portugal” to “detected in Portugal”

Author Response

To begin with, we would like to thank the editorial board of PATHOGENS and the three reviewers who have reviewed our original work for their comments/suggestions. Direct replies to the comments made in the reviewing process can be found below. As suggested, we have (extensively) modified the original text of our manuscript and corrected it for typos and introduced modifications that we belied will answer to the all the points raised by the 3 reviewers engaged for its analysis.

We hope to have been able to prepare a modified version of our work that is clearer than the previous one, and also expect to have correctly addressed all the remarks raized by the reviewers. The corrected version of our manuscript has been uploaded to the MDPI submission portal as pathogens-2111196_corrected. In addition, figure 2 has now been presented as supplementary data, Table 1 has been modified, and all figures have been processed to have 300 dpi.

Reviewer_1

Comments and Suggestions for Authors

This manuscript by Marta Falcão et al, aimed to report the Genome Characterization and Spaciotemporal Dispersal Analysis of Bagaza Virus detected in Portugal, 2021. Phylogenies inferred from nucleotide, complemented with the analysis of amino 23 acid alignments, indicated that this strain from Portugal is closely related to BAGV strains previously detected in Spain (2010 and 2019). The authors also argued that there is a crucial need to set up surveillance of BAGV and associated vectors to allow more effective and rapid control of the virus, mitigating the implication on public and animal health. 

Please, the following comments should be considered for revisions to the manuscript:

Major revisions

The clarity of the captions should be improved to 300 dpi. They are not readable in the current version.

Reply: new, 300dpi figures were included in the text. Figure 2 has now been changed to supplementary data.

The comparative analysis performed in this study didn’t include the 5’ and 3’ NCR that play an important role in Bagaza virus diversity. Could the author assess the polymorphism in these regions at least between the BAGV strains from Spain and Portugal?

Reply: We analysed the available sequence for both the 5' and 3'-UTRs, and the results from both these analyses were added to the text.

Lines 304 and 370, the authors have written “probably” which mean that the estimates did not support this statement. Could they provide more details regarding this uncertainty (line 376-380)? I recommend that the authors performed a regressed root-to-tip analysis including only BAGV sequences to ensure sufficient time structures in alignment for reliable rate estimates, determine accurate sampling dates and perform again a separated MCMC for BAGV sequences.

Reply: as suggested the figure shows the root-to-tip analysis including BAGV/ITV sequences, indicating an acceptable temporal signal (r2=0.35).

The same holds true with the TMUV subdataset, for which a 0.29 r2 value was obtained.

The text has been slightly changed to try and make it a little bit more clear.

Reply: a new section has been added to the text, where results from the UTR sequence comparisons are now reported.

The abstract is very poorly written and could be improved the more details summarizing the findings. The discussion could be improved by comparing your findings to previous studies.

Reply: both the abstract and the discussion have been changed in an attempt to improve them.

Minor revisions:

Line 42, could the authors change “the so-called the non-structural proteins” to “the so-called non-structural proteins”?

Reply: the text has been corrected as suggested.

Line 54, could the authors change “benn made” to “been made”?

Reply: the text has been corrected as suggested.

Lines 170-171, could the authors rephrase for more clarity?

Reply: several changes have been introduced in the text to try and make it as clear as possible.

Lines 214-216, could the authors rephrase for more clarity?

Reply: several changes have been introduced in the text to try and make it as clear as possible.

Line 261, could the authors change “lienage” to “lineage”?

Reply: the text has been corrected as suggested.

Lines 285-291 seems to be a duplicate of a section in the Methods. It could be rephrased or moved to the methods section and avoid mixing methods with results.

Reply: several changes have been introduced in the text to try and make it as clear as possible.

Lines 311-318 should be moved before lines 293-310. The authors should describe the MCC from 3a to 3c.

Reply: the text has been corrected as suggested.

Line 339, change “detected i Portugal” to “detected in Portugal”

Reply: the text has been corrected as suggested.

Reviewer 2 Report

Manuscript ID: pathogens-2111196

Title: Genome Characterization and Spaciotemporal Dispersal Analysis of Bagaza Virus detected in Portugal, 2021

In this manuscript,  Falcão et al. report on the phylogenetic characterization of  Bagaza virus (BAGV), a member of the Ntaya group, within the Flavivirus genus. The focus is on genomic characterization of BAGV and its relatedness with other flaviviruses. Furthermore, authors suggest that  BAGV and Israel turkey meningoencephalitis virus (ITV) belong to the same species since they consistently group together in one single cluster.

In general, this manuscript is an interesting subject. The research method is well thought, and the interpretation is adequate and in accord with their results.

However I have several issues in the manuscript and I would like to address the Authors

METHODS

line 106   Please clarify why you mentioned carcasses of two red-legged partridges, if you used tissue samples only from one? Did you analyse the samples of the other one?

Table 1: Please, put the annealing temperature of each primer par in the Table 1

Please refer the lengths of alignment used for phylogenetic analysis in the section 2.4.

Did you analyse your sequence dataset in order to detect potential recombination? Recombinants must be excluded from in-depth phylogenetic analysis

Line 180: correct a typo (burn-in instead of buri-in)

line 186:  You can add abbreviation- SPREAD

RESULTS

line 203 correct a typo (when instead of whem)

line 216 Reword the sentence to make the thoughts clear

Line 275 correct a typo

Line 280 correct a typo

Figure 3 Figure is not clear enough, please resize it

Author Response

To begin with, we would like to thank the editorial board of PATHOGENS and the three reviewers who have reviewed our original work for their comments/suggestions. Direct replies to the comments made in the reviewing process can be found below. As suggested, we have (extensively) modified the original text of our manuscript and corrected it for typos and introduced modifications that we belied will answer to the all the points raised by the 3 reviewers engaged for its analysis.

We hope to have been able to prepare a modified version of our work that is clearer than the previous one, and also expect to have correctly addressed all the remarks raized by the reviewers. The corrected version of our manuscript has been uploaded to the MDPI submission portal as pathogens-2111196_corrected. In addition, figure 2 has now been presented as supplementary data, Table 1 has been modified, and all figures have been processed to have 300 dpi.

Revier_2

Comments and Suggestions for Authors

Manuscript ID: pathogens-2111196

Title: Genome Characterization and Spaciotemporal Dispersal Analysis of Bagaza Virus detected in Portugal, 2021

In this manuscript,  Falcão et al. report on the phylogenetic characterization of  Bagaza virus (BAGV), a member of the Ntaya group, within the Flavivirus genus. The focus is on genomic characterization of BAGV and its relatedness with other flaviviruses. Furthermore, authors suggest that  BAGV and Israel turkey meningoencephalitis virus (ITV) belong to the same species since they consistently group together in one single cluster.

In general, this manuscript is an interesting subject. The research method is well thought, and the interpretation is adequate and in accord with their results.

However I have several issues in the manuscript and I would like to address the Authors

METHODS

line 106   Please clarify why you mentioned carcasses of two red-legged partridges, if you used tissue samples only from one? Did you analyse the samples of the other one?

Reply: several changes have been introduced in the text to try and make it as clear as possible.

Table 1: Please, put the annealing temperature of each primer par in the Table 1

Reply: several changes have been introduced in table in order to comply with the suggestion made.

Please refer the lengths of alignment used for phylogenetic analysis in the section 2.4.

Reply: several changes have been introduced in the text to comply with the suggestion made.

Did you analyse your sequence dataset in order to detect potential recombination? Recombinants must be excluded from in-depth phylogenetic analysis

Reply: the sequence alignments were analysed for signs of recombination using RDP4 but none were found. This information was added to the text (first paragraph, section 2.4).

Line 180: correct a typo (burn-in instead of buri-in)

Reply: the text has been corrected.

line 186:  You can add abbreviation- SPREAD

Reply: the text has been corrected.

RESULTS

line 203 correct a typo (when instead of whem)

Reply: the typo in question has been corrected.

line 216 Reword the sentence to make the thoughts clear

Reply: the text has been changed, also taking into account the same comment made by reviewer 1.

Line 275 correct a typo

Reply: the typo in question has been corrected.

Line 280 correct a typo

Reply: the typo in question has been corrected.

Figure 3 Figure is not clear enough, please resize it

Reply: Figure 2 has now been added as supplementary data. All figures were processed to have 300 dpi.

Reviewer 3 Report

The manuscript “Genome Characterization and Spaciotemporal Dispersal Analysis of Bagaza Virus detected in Portugal, 2021” by Falcao et al. reports the genomic characterization of the BAGV strain detected in Portugal last year (Queiros et al., 2022, reference 17). Phylogenetic analysis reveals a close homology with BAGV isolates from Spain.

The study includes relevant data about epidemiology of WNV in the Iberian Peninsula suggesting an endemicity in the area and a transmission from Spain to Portugal.

Nevertheless, some conclusions obtained in the analysis of the data are not sufficiently supported and other cannot be considered a novelty. The conclusion described in the article indicating that BAGV and Israel turkey meningoencephalitis virus (ITV) are the same viral species has been previously published by Pinero et al., (reference 3 in the manuscript). In that publication Pinero et al. obtain the complete sequence of 5 ITV strain and compare with phylogenetic analysis with BAGV strains determining that all of them belong to the same clade, and that genetic distances between BAGV and ITV are in the same range as when you analyze only BAGV or ITV (as you also describe in the manuscript). Pinero et al. also determine the divergences in polyproteins in BAGV and ITV strains detecting a high homology (less than 1.5% of divergence). Consequently, although new analysis as PCOORD and Neighbour-net network have been included in this manuscript, the finding that BAGV and ITV are the same viral species has been unequivocally described, and it cannot be considered as original research, and I consider that that analysis and conclusions needs to be removed from the manuscript, and consequently specific phylogenetic analysis to confirm that (PCCORD and Neighbour-net network are not required. Although they have not been officially disclosed as member of the same viral species by ICTV, it does not change the fact that it has been previously published.

Other major issue observed in the manuscript are the conclusions obtained by spaciotemporal analyses. As you mention in the discussion the transcontinental spread of flaviviruses is frequently associated with bird migrations and probably with massive tourism and commercial trading. All these situations can move a specific strain in a fast way in sporadic introductions. Also, as you explain also in the discussion the number of sequences available is low and are highly heterogeneous, leading to errors in timing and geographic dispersal events deduced from spatiotemporal analysis. Also, a problem in the sequences from Spain used for the analysis has been detected and it could affect the result of the analyses. The six BAGV sequences from Spain 2012 were obtained from the same animal: HQ644144 and HQ644143 were obtained from the brain and the heart from the bird, and KR108244, KR108245 and KR108244 were obtained from the heart of the animal after different cell passages. Consequently, all these sequences should be considered the same virus and should be included only one these sequences in the analysis. Considering this problem, you should repeat the phylogenetic analysis with only one of the sequences and probably it could affect the results of spaciotemporal analysis. Considering the next facts: this virus could be transported in a rapid way from one place to other, there are a low number of sequences analysed (with only one real sequence from Spain and one from Portugal) I consider that the conclusion of the introduction in the Iberian Peninsula in 2003 is not consistently supported by the data obtained. Although, after de correct analysis, you obtain a result that indicates a common ancestor in 2003, it seems highly probable that the virus could be introduced in Spain between 2003 and 2010 (first detection in Spain). In my opinion, the suggestion that BAGV arrived in the Iberian Peninsula is not consistently supported and is too risky and I consider that it should be eliminated from the abstract and consider that it could have arrived later.

Other issues:

Figures are very small, it his highly difficult to read the number of the sequences in Figure and and completely impossible in Figures 2 and 3

Abstract: line 25. It is described that Portugal strain is related to Spanish BAGV strains from 2010 and 2019, nevertheless 2019 sequences have not been included in the study, and consequently it should be eliminated from the sentence. 2019 strain has been sequenced only partially (Höfle et al., 2022, reference 11) and interestingly this short sequence is more similar to those from Senegal that to the previous sequences from Spain. In that work the authors also include other short sequence from 2013 that almost identical to 2010 strain. I suggest not to include these data considering the short length of the sequences.

Introduction: Lines 80-83. Include in that paragraph the reference Gamino et al., (reference 17) where the presence of BAGV in pigeons is detected.

Materials and methods:

In section 2.1 (The Study) you describe that heart and tissue sample from partridge number 7 were used for molecular characterization and virus isolation. Nevertheless, in the next section (2.2, RNA extraction and viral RNA detection) it is described that nucleic acid was extracted from heart, brain, kidney and intestine from two animals, but these materials have not been used in the study. Please delete the information about tissues not used.

Section 2.5: I suggest eliminate PCOORD and phylogenetic neighbour-net networks considering to determine ITV and BAGV relationship, because this fact has been previously published.

Results:

Line 213: change “whem” to “when”

Line 217: After “were found in BAGVPT/2021” add “when compared to…(the other sequences)”

Line 259-256: When you write “versus southern Africa (Namibia/Zambia) 259 versus Iberia (Spain/Portugal)” I think that you men “versus southern Africa (Namibia/Zambia) and Iberia (Spain/Portugal)”

Discussion:

Lines 389-391: I cannot understand clearly the sentence. Probably you should talk about viral species instead genetic lineages, that is mostly used as variations into a viral species

Data Availability Statement (lines 426 and 427): You talk about “seven E33 strains” detected in the study uploaded to Genbankc but you only include one sequence (LC730845) corresponding to BAGV Portugal/2021

Author Response

To begin with, we would like to thank the editorial board of PATHOGENS and the three reviewers who have reviewed our original work for their comments/suggestions. Direct replies to the comments made in the reviewing process can be found below. As suggested, we have (extensively) modified the original text of our manuscript and corrected it for typos and introduced modifications that we belied will answer to the all the points raised by the 3 reviewers engaged for its analysis.

We hope to have been able to prepare a modified version of our work that is clearer than the previous one, and also expect to have correctly addressed all the remarks raized by the reviewers. The corrected version of our manuscript has been uploaded to the MDPI submission portal as pathogens-2111196_corrected. In addition, figure 2 has now been presented as supplementary data, Table 1 has been modified, and all figures have been processed to have 300 dpi.

Reviewer_3

Comments and Suggestions for Authors

The manuscript “Genome Characterization and Spaciotemporal Dispersal Analysis of Bagaza Virus detected in Portugal, 2021” by Falcao et al. reports the genomic characterization of the BAGV strain detected in Portugal last year (Queiros et al., 2022, reference 17). Phylogenetic analysis reveals a close homology with BAGV isolates from Spain. 

The study includes relevant data about epidemiology of WNV in the Iberian Peninsula suggesting an endemicity in the area and a transmission from Spain to Portugal.

Nevertheless, some conclusions obtained in the analysis of the data are not sufficiently supported and other cannot be considered a novelty. The conclusion described in the article indicating that BAGV and Israel turkey meningoencephalitis virus (ITV) are the same viral species has been previously published by Pinero et al., (reference 3 in the manuscript). In that publication Pinero et al. obtain the complete sequence of 5 ITV strain and compare with phylogenetic analysis with BAGV strains determining that all of them belong to the same clade, and that genetic distances between BAGV and ITV are in the same range as when you analyze only BAGV or ITV (as you also describe in the manuscript). Pinero et al. also determine the divergences in polyproteins in BAGV and ITV strains detecting a high homology (less than 1.5% of divergence). Consequently, although new analysis as PCOORD and Neighbour-net network have been included in this manuscript, the finding that BAGV and ITV are the same viral species has been unequivocally described, and it cannot be considered as original research, and I consider that that analysis and conclusions needs to be removed from the manuscript, and consequently specific phylogenetic analysis to confirm that (PCCORD and Neighbour-net network are not required. Although they have not been officially disclosed as member of the same viral species by ICTV, it does not change the fact that it has been previously published. 

Reply: it was not our intention to appropriate the suggestion that BAGV and ITV correspond to different strains of the same viral species. Indeed, the reference proposing the latter has always been part of this manuscript reference list. On the other hand, we understand the referee's comment. What we intended to do was to give additional support to the proposal previously made by Fernández-Pinero and collaborators, since the use of analyses that were different than those previously used, came to a similar result. Accordingly, the text has not only been changed so as to make our point clearer, but our data have also been removed from the main body of text, and added as supplementary information.

Other major issue observed in the manuscript are the conclusions obtained by spaciotemporal analyses. As you mention in the discussion the transcontinental spread of flaviviruses is frequently associated with bird migrations and probably with massive tourism and commercial trading. All these situations can move a specific strain in a fast way in sporadic introductions. Also, as you explain also in the discussion the number of sequences available is low and are highly heterogeneous, leading to errors in timing and geographic dispersal events deduced from spatiotemporal analysis. Also, a problem in the sequences from Spain used for the analysis has been detected and it could affect the result of the analyses. The six BAGV sequences from Spain 2012 were obtained from the same animal: HQ644144 and HQ644143 were obtained from the brain and the heart from the bird, and KR108244, KR108245 and KR108244 were obtained from the heart of the animal after different cell passages. Consequently, all these sequences should be considered the same virus and should be included only one these sequences in the analysis. Considering this problem, you should repeat the phylogenetic analysis with only one of the sequences and probably it could affect the results of spaciotemporal analysis. Considering the next facts: this virus could be transported in a rapid way from one place to other, there are a low number of sequences analysed (with only one real sequence from Spain and one from Portugal) I consider that the conclusion of the introduction in the Iberian Peninsula in 2003 is not consistently supported by the data obtained. Although, after de correct analysis, you obtain a result that indicates a common ancestor in 2003, it seems highly probable that the virus could be introduced in Spain between 2003 and 2010 (first detection in Spain). In my opinion, the suggestion that BAGV arrived in the Iberian Peninsula is not consistently supported and is too risky and I consider that it should be eliminated from the abstract and consider that it could have arrived later.

Reply: due to lack of time to perform the suggested analysis, we decided to redo the phylogenetic reconstruction using a dataset with only BAGV/ITV sequences, from which 100% identical sequences were removed with ElimDupes (https://www.hiv.lanl.gov/content/sequence/elimdupesv2/elimdupes.html). Two sequences were removed (KR108246_Bagaza_virus_Spain_2020 AND HQ644144_Bagaza_virus_B_Spain_2010). Unfortunately, the obtained tree (see below) was less probable than the one previously presented, including the internal position of the branch placing the obtained BAGV sequence (from Portugal). Again, an African origin was suggested for the subcluster that includes the Portuguese and Spanish BAGV (though at locations that were slightly different from our previous analysis), the latter (spanish cluster) sharing an apparent ancestor around 11 ago (as the previous analysis had suggested). Therefore, while an African origin for the BAGV from Iberia is still indicated, exactly when their shared common ancestor occurred in a recent time requires further analyses with larger datasets, both in terms of sampling dates and geographic coverage. Therefore, the text has been changed so as not to indicate a specific date for the Iberian BAGV cluster.

Other issues:

Figures are very small, it his highly difficult to read the number of the sequences in Figure and and completely impossible in Figures 2 and 3

Reply: new figures have been prepared and added to the text.

Abstract: line 25. It is described that Portugal strain is related to Spanish BAGV strains from 2010 and 2019, nevertheless 2019 sequences have not been included in the study, and consequently it should be eliminated from the sentence. 2019 strain has been sequenced only partially (Höfle et al., 2022, reference 11) and interestingly this short sequence is more similar to those from Senegal that to the previous sequences from Spain. In that work the authors also include other short sequence from 2013 that almost identical to 2010 strain. I suggest not to include these data considering the short length of the sequences.

Reply: the text has been changed, also as a result of other suggestions made by other reviewers. Our analysis was exclusively based on viral sequences for which a complete ORF was available.

Introduction: Lines 80-83. Include in that paragraph the reference Gamino et al., (reference 17) where the presence of BAGV in pigeons is detected.

Reply: the text has been changed accordingly.

Materials and methods:

In section 2.1 (The Study) you describe that heart and tissue sample from partridge number 7 were used for molecular characterization and virus isolation. Nevertheless, in the next section (2.2, RNA extraction and viral RNA detection) it is described that nucleic acid was extracted from heart, brain, kidney and intestine from two animals, but these materials have not been used in the study. Please delete the information about tissues not used.

Reply: slight changes introduced in the text have tried to make this text section a little clearer.

Section 2.5: I suggest eliminate PCOORD and phylogenetic neighbour-net networks considering to determine ITV and BAGV relationship, because this fact has been previously published.

Reply: we understand the argument but would, nonetheless, mention the support we obtained for BAGV/ITV being a single species, but since this observation has been published before, and is not entirely original (although confirmed with different approaches to those used by Fernándes-Pinero), we changed fig_2 (PCOORD/NNn) to supplementary data.

Results:

Line 213: change “whem” to “when”

Reply: the text has been corrected.

Line 217: After “were found in BAGVPT/2021” add “when compared to…(the other sequences)”

Reply: the text has been changed to make it as clear as possible.

Line 259-256: When you write “versus southern Africa (Namibia/Zambia) 259 versus Iberia (Spain/Portugal)” I think that you men “versus southern Africa (Namibia/Zambia) and Iberia (Spain/Portugal)”

Reply: The text has been corrected

Discussion:

Lines 389-391: I cannot understand clearly the sentence. Probably you should talk about viral species instead genetic lineages, that is mostly used as variations into a viral species

Reply: the text has been corrected to make it clearer

Data Availability Statement (lines 426 and 427): You talk about “seven E33 strains” detected in the study uploaded to Genbankc but you only include one sequence (LC730845) corresponding to BAGV Portugal/2021

Reply: we apologise for the mistake. The text has been corrected

Round 2

Reviewer 3 Report

The new version of the manuscript has improved a high number of issues that have been suggested by the reviewers. Nevertheless, one of the main problems has not been solved.

As I suggested in the previous review, you should include only one sequence from BAGV from Spain 2022. The reason was that all the 6 sequences included were obtained from the same animal, and consequently all of them correspond to an only natural strain. For a phylogenetic analysis and a spaciotemporal study only natural variants should be included. Two of the sequences included in the phylogenetic trees were obtained from the brain and the heart in the same partridge, and the other 3 sequences were obtained after different cell passage post-isolation from heart. In your author answers you explain that when you remove two of these sequences (because they were identical to others), the obtained tree was less probable and the determination of the time of occurrence of the common ancestor and the place of origin is not precise. When the 5 sequences of BAGV Spain are aligned we can see that only 3 nucleotides show differences, and in all the cases these differences are present in the viral sequences obtained after cell passages (KR108244, KR108245 and KR108246) in relation with the sequence of the original virus obtained directly from heart macerate. Heart and brain sequence from macerates are also identical). It is clear that the changes observed have been caused during cell culture. Of course, for a study of natural variants in the nature, only the original sequence, and not sequence with changes induced by culture must be used. Consequently, you have only a sequence from Spain 2010 that can be used in Phylogenetic studies. The inclusion of other not valid sequence will disturb the results in a false way, as indicates the fact that the reduction from 5 to 3 sequences affects the results of the tree.

I consider that the “lack of time” is not a valid reason to include the correct analysis in the manuscript, as you explain in the authors response. You should perform the correct analysis by using only one sequence from Spain 2020 (preferably HQ644143 from heart, because it has been used in other analysis in the manuscript) and use these analyses and the new figures obtained in the manuscript. I consider that if you include more than one sequence from the same virus the result will be incorrectly modified, and the conclusions obtained will not be valid. Consequently, if this modification is not applied, I will recommend that the manuscript will be rejected. When the analysis with only one Spanish sequence will be done, probably the determination of the date and place where a common ancestor existed will not be statistically supported and the proposal of a date of the presence in the Iberian Peninsula will not be supported by the data. Consequently, probably you can´t propose a date for the introduction in Spain. Consequently, the contribution to the spaciotemporal analysis for BAGV strain from Portugal is not as relevant as expected in the first version of the manuscript. By this reason I suggest eliminating the “Spaciotemporal dispersal analysis” from the title of the article, that now could be similar to: “Genome Characterization and Phylogenetic analysis of Bagaza Virus detected in Portugal, 2021”

Other issues:

Supplementary figures 1 and 2 have not been uploaded to the web and can´t be revised.

The size of the figures 1 and 2 are still too small (mainly figure 2, where even using the zoom in the computer it is impossible to read correctly)

The suggestion from my first revision in relation to Spain 2019 strain has not been followed. You confirm that you only use sequences with complete ORF available, but you don´t have eliminated 2109 strain from line 25 in the abstract.

I you perform the new analysis suggested previously, you probably will need to eliminate from the abstract that the common ancestor have arrived at the Iberian Peninsula in the early 2000s if it is not supported statistically in the phylogenetic analysis.

You have not followed the next suggestion from the first revision:

“Introduction: lines 80-83. Include in that paragraph the reference Gamino et al., (reference 17) where the presence of BAGV in pigeons is detected”. You should change reference 7 to 17 after “…to a lesser extent, in common wood pigeons”

In line81 in the introduction the reference 5 is not correctly used, because that article is focused in mosquito vectors and not in mortality in birds (The correct references are 9, 12 and 17)

Other minor issues:

Materials and methods: line 115: change “partridges” to “partridge”

Lines 123-124: delete “from partridge no7, which showed the lower Ct values (20.4 and 21.52, respectively”. It has been explained before that the partridge was selected by the low Ct values

Line 178: change “Boststrap” to “Bootstrap”

Results:

Lines 222-225: The sentence has been modified but it is still highly confusing. You have previously explained that capsid protein from BAGV from Portugal is the protein with the lowest similarity with BAGV from Spain and ITV and you don´t need to repeat the data. Additionally, it is false that capsid protein is the most different in relation with NTAV and TMUV where the most divergent protein is NS2A. I suggest delete the sentence, because it is not relevant.

Lines 297-298: Reference to Table 1 from supplementary material is not correctly used, because in the table does not indicate the small sequence dataset. I consider that the reference is not required, because later you indicate that the dataset is comprised of all BAGV/ITV, TMUV and NTAV sequences.

Lines 319-321: As described previously and supported in this work BAGV and ITV are the same viral species. Consequently, there is not point in talking about a common ancestor of BAGV and ITV because they are the same virus and belong to the same clade. You could say that “the common ancestor of BAGV/ITV virus strains has been determined in North-Sudan….”

Lines 325-327: The data of possible introduction in the Iberian Peninsula must be modified or eliminated after new phylogenetic analysis.

Discussion:

Line 374: Change “NATV” to “NTAV”

Line375: Delete add

Line 393: Delete recent

Author Response

The new version of the manuscript has improved a high number of issues that have been suggested by the reviewers. Nevertheless, one of the main problems has not been solved. 

As I suggested in the previous review, you should include only one sequence from BAGV from Spain 2022. The reason was that all the 6 sequences included were obtained from the same animal, and consequently all of them correspond to an only natural strain. For a phylogenetic analysis and a spaciotemporal study only natural variants should be included. Two of the sequences included in the phylogenetic trees were obtained from the brain and the heart in the same partridge, and the other 3 sequences were obtained after different cell passage post-isolation from heart. In your author answers you explain that when you remove two of these sequences (because they were identical to others), the obtained tree was less probable and the determination of the time of occurrence of the common ancestor and the place of origin is not precise. When the 5 sequences of BAGV Spain are aligned we can see that only 3 nucleotides show differences, and in all the cases these differences are present in the viral sequences obtained after cell passages (KR108244, KR108245 and KR108246) in relation with the sequence of the original virus obtained directly from heart macerate. Heart and brain sequence from macerates are also identical). It is clear that the changes observed have been caused during cell culture. Of course, for a study of natural variants in the nature, only the original sequence, and not sequence with changes induced by culture must be used. Consequently, you have only a sequence from Spain 2010 that can be used in Phylogenetic studies. The inclusion of other not valid sequence will disturb the results in a false way, as indicates the fact that the reduction from 5 to 3 sequences affects the results of the tree.

I consider that the “lack of time” is not a valid reason to include the correct analysis in the manuscript, as you explain in the authors response. You should perform the correct analysis by using only one sequence from Spain 2020 (preferably HQ644143 from heart, because it has been used in other analysis in the manuscript) and use these analyses and the new figures obtained in the manuscript. I consider that if you include more than one sequence from the same virus the result will be incorrectly modified, and the conclusions obtained will not be valid. Consequently, if this modification is not applied, I will recommend that the manuscript will be rejected. When the analysis with only one Spanish sequence will be done, probably the determination of the date and place where a common ancestor existed will not be statistically supported and the proposal of a date of the presence in the Iberian Peninsula will not be supported by the data. Consequently, probably you can´t propose a date for the introduction in Spain. Consequently, the contribution to the spaciotemporal analysis for BAGV strain from Portugal is not as relevant as expected in the first version of the manuscript. By this reason I suggest eliminating the “Spaciotemporal dispersal analysis” from the title of the article, that now could be similar to: “Genome Characterization and Phylogenetic analysis of Bagaza Virus detected in Portugal, 2021”

Reply: the deletion of the indicated sequences from the dataset resulted in changes especially in the age of the MRCA, as the date indicated is associated with a large 96% HPD. However, under this situation, the date for the ancestor of the BAGV detected in Spain and Portugal is relatively recent, and was suggested to have existed 46 years ago (probable location in Spain). These results are presented below. Dates and locations are indicated for the branches closest to the iberian BAGV sequences. As changes in the used dataset changes the timing of splitting events probably due to limited sampling the text has been changed to warn the reader about its consequences (lines 428-432, 554-558).

Other issues:

Supplementary figures 1 and 2 have not been uploaded to the web and can´t be revised.

Reply: supplementary material has been uploaded

The size of the figures 1 and 2 are still too small (mainly figure 2, where even using the zoom in the computer it is impossible to read correctly)

Reply: We tried to increase the resolution of the images to help their analysis

The suggestion from my first revision in relation to Spain 2019 strain has not been followed. You confirm that you only use sequences with complete ORF available, but you don´t have eliminated 2109 strain from line 25 in the abstract.

Reply: the text has been corrected

I you perform the new analysis suggested previously, you probably will need to eliminate from the abstract that the common ancestor have arrived at the Iberian Peninsula in the early 2000s if it is not supported statistically in the phylogenetic analysis.

Reply: the text has been corrected to accommodate for uncertainties in the dating of events (as a result of new analysis)

You have not followed the next suggestion from the first revision:

“Introduction: lines 80-83. Include in that paragraph the reference Gamino et al., (reference 17) where the presence of BAGV in pigeons is detected”. You should change reference 7 to 17 after “…to a lesser extent, in common wood pigeons”

Reply: the text has been corrected

In line81 in the introduction the reference 5 is not correctly used, because that article is focused in mosquito vectors and not in mortality in birds (The correct references are 9, 12 and 17)

Reply: the text has been corrected

Other minor issues:

Materials and methods: line 115: change “partridges” to “partridge”

Reply: the text has been corrected

Lines 123-124: delete “from partridge no7, which showed the lower Ct values (20.4 and 21.52, respectively”. It has been explained before that the partridge was selected by the low Ct values

Reply: the text has been corrected

Line 178: change “Boststrap” to “Bootstrap”

Reply: the text has been corrected

Results:

Lines 222-225: The sentence has been modified but it is still highly confusing. You have previously explained that capsid protein from BAGV from Portugal is the protein with the lowest similarity with BAGV from Spain and ITV and you don´t need to repeat the data. Additionally, it is false that capsid protein is the most different in relation with NTAV and TMUV where the most divergent protein is NS2A. I suggest delete the sentence, because it is not relevant.

Reply: the text has been corrected

Lines 297-298: Reference to Table 1 from supplementary material is not correctly used, because in the table does not indicate the small sequence dataset. I consider that the reference is not required, because later you indicate that the dataset is comprised of all BAGV/ITV, TMUV and NTAV sequences.

Reply: the text has been corrected

Lines 319-321: As described previously and supported in this work BAGV and ITV are the same viral species. Consequently, there is not point in talking about a common ancestor of BAGV and ITV because they are the same virus and belong to the same clade. You could say that “the common ancestor of BAGV/ITV virus strains has been determined in North-Sudan….”

Reply: the text has been corrected

Lines 325-327: The data of possible introduction in the Iberian Peninsula must be modified or eliminated after new phylogenetic analysis.

Reply: the text has been corrected

Discussion: 

Line 374: Change “NATV” to “NTAV”

Reply: the text has been corrected

Line375: Delete add

Reply: the text has been corrected

Line 393: Delete recent

Reply: the text has been corrected
